# Curve your Enthusiasm: Concurvity Regularization in Differentiable Generalized Additive Models

Julien Siems [* 1]   Konstantin Ditschuneit [* 1]   Winfried Ripken [* 1]   Alma Lindborg [* 1]   Maximilian Schambach [1]
Johannes S. Otterbach [2]   Martin Genzel [1]

## Abstract

Generalized Additive Models (GAMs) have recently experienced a resurgence in popularity, particularly in high-stakes domains such as healthcare. GAMs are favored due to their interpretability, which arises from expressing the target value as a sum of non-linear functions of the predictors. Despite the current enthusiasm for GAMs, their susceptibility to concurvity – i.e., (possibly non-linear) dependencies between the predictors – has hitherto been largely overlooked. Here, we demonstrate how concurvity can severely impair the interpretability of GAMs and propose a remedy: a conceptually simple, yet effective regularizer which penalizes pairwise correlations of the non-linearly transformed feature variables. This procedure is applicable to any gradient-based fitting of differentiable additive models, such as Neural Additive Models or NeuralProphet, and enhances interpretability by eliminating ambiguities due to self-canceling feature contributions. We validate the effectiveness of our regularizer in experiments on synthetic as well as real-world datasets for time-series and tabular data. Our experiments show that concurvity in GAMs can be reduced without significantly compromising prediction quality, improving interpretability and reducing variance in the feature importances.

## 1. Introduction

Interpretability has emerged as a critical requirement of machine learning models in domains involving high-stakes decisions and regulatory constraints, such as healthcare. In this domain, it is of vital importance to ensure transparency and accountability of decisions (e.g., for or against medical intervention) as well as to ensure that biases and confounding effects in existing data are well-understood (Caruana et al., 2015; Zhang et al., 2022; Chang et al., 2021). Similar concerns affect applications such as loan approvals (Arun et al., 2016) and hiring practices (Dattner et al., 2019), among others (Barocas & Selbst, 2016; Rudin et al., 2022). In the healthcare sector in particular, the clarity of interpretation can often be a major hurdle in securing approval for medical devices from medical authorities like the FDA or EMA, because doctors are required to justify their decisions. Therefore, the ability to clearly interpret a model might be given priority over its predictive accuracy, as described in Letham et al. (2015)

A popular model class for interpretable machine learning is *Generalized Additive Models* (*GAMs*) (Hastie & Tibshirani, 1987), in which the target variable is expressed as a sum of non-linearly transformed features. GAMs combine the interpretability of (generalized) linear models with the flexibility to capture non-linear dependencies between the features and the target. GAMs have recently seen a resurgence in interest with prominent examples being *Neural Additive Models* (*NAMs*) (Agarwal et al., 2021) and its variants (Chang et al., 2022; Dubey et al., 2022; Radenovic et al., 2022; Xu et al., 2022; Enouen & Liu, 2022) for tabular data, as well as *Prophet* (Taylor & Letham, 2018) and *NeuralProphet* (Triebe et al., 2021) for time-series forecasting. Both domains will be further explored in our experiments.

A significant obstacle to the interpretability of additive models is the phenomenon of *concurvity* (Buja et al., 1989). As a non-linear analog to multicollinearity, concurvity refers to the presence of strong correlations among the non-linearly transformed feature variables. Similarly to multicollinearity, concurvity can impair interpretability because parameter estimates become unstable when features are correlated (Ramsay et al., 2003), resulting in highly disparate interpretations of the data depending on the model initialization. Although this issue is known and addressed by various techniques such as variable selection (Kovács, 2022) in traditional GAMs, it has been overlooked in more recent works. Unlike the prevalent GAM package *mgcv* (Wood, 2001), we are not aware of any differentiable GAM implementations

*Equal contribution [1]Merantix Momentum [2]Work done while at Merantix Momentum. Correspondence to: Julien Siems <juliensiems@gmail.com>.

*Workshop on Interpretable ML in Healthcare at International Conference on Machine Learning (ICML)*, Honolulu, Hawaii, USA. 2023. Copyright 2023 by the author(s).

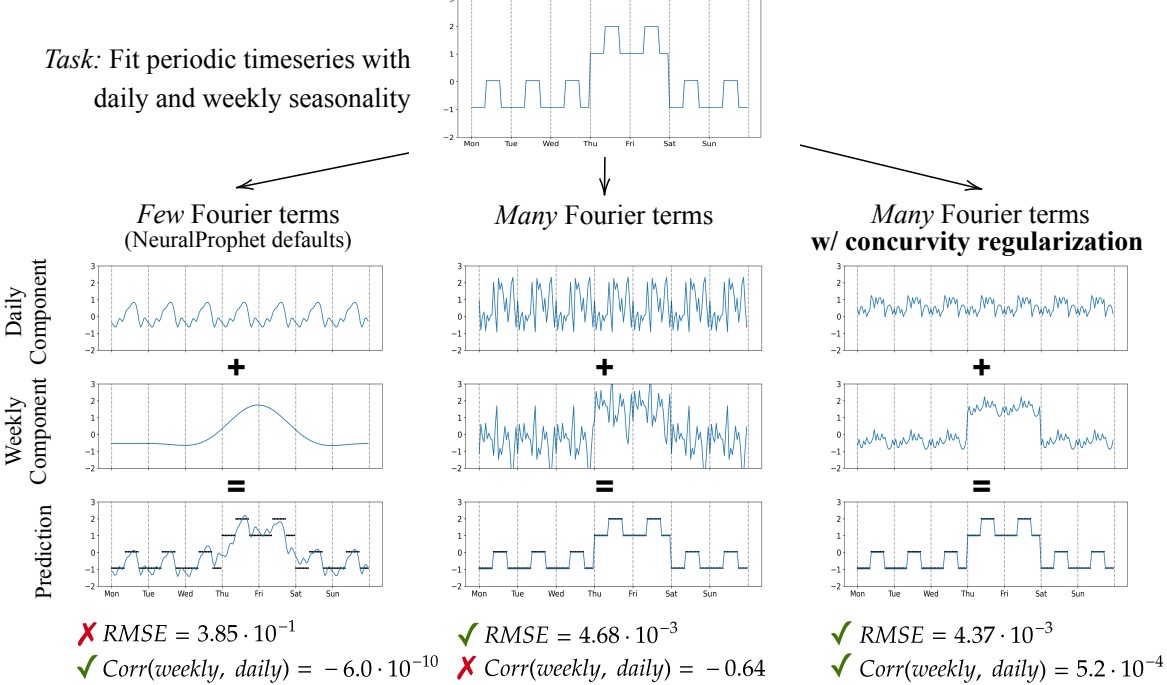

Figure 1. Concurvity in a NeuralProphet model: Fitting a time series composed of daily and weekly seasonalities, each represented by Fourier terms. (left) Using few Fourier terms results in uncorrelated components but a poor fit; (middle) A more complex model improves the fit but sacrifices interpretability due to self-canceling high-frequency terms; (right) The same complex model, but with our *regularizer*, achieves both good predictive performance and interpretable (decorrelated) components. See Appendix D.2 for details.

that include concurvity metrics.

In this work, we propose a novel regularizer for reducing concurvity in GAMs by penalizing pairwise correlations of the non-linearly transformed features. Reducing concurvity improves interpretability because it promotes the isolation of feature contributions to the target by eliminating potentially correlated or self-canceling transformed feature combinations. As a result, the model becomes easier to inspect by clearly separating individual feature contributions. In addition, our regularizer encourages the model to learn more consistent feature importances across model initializations, which increases interpretability. The trade-off between increased interpretability and prediction quality will be further explored in Section 4.

The example depicted in Figure 1 provides a first intuition of the practical implications of concurvity and how these can be addressed by our regularizer. We use the additive time series model NeuralProphet (Triebe et al., 2021). For this particular example, we have limited the model to incorporate only daily and weekly seasonality components. These components are modeled using periodic functions whose complexity can be adjusted as needed. More details on the experiment can be found in Appendix D.2. We find that while the default NeuralProphet parameters effectively mitigate concurvity by producing a very simple model, they

provide a worse fit to the data than the more complex models. However, if left unregularized, a more complex model is subject to strong correlation between the seasonalities, an effect visually apparent in self-canceling periodic components in the middle column of Figure 1. In contrast, when using our regularization, the seasonalities are less correlated, resulting in a clearer separation between the components. While the predictive performance of the two complex models is comparable, the regularized model is more interpretable because daily and weekly effects are clearly separated. We argue that in general, a GAM with lower concurvity is preferable to a GAM with similar prediction quality but higher concurvity.

Our main contributions are:

1. We showcase the susceptibility of modern additive models to concurvity and present a revised formal definition of the concept.

2. We propose a concurvity regularizer applicable to any differentiable GAM.

3. We validate our approach experimentally using a variety of synthetic as well as real-world data, investigating the trade-off between concurvity and prediction quality, as well as the impact of regularization on interpretability.

## 2. Background

### 2.1. Generalized Additive Models

*Generalized Additive Models* (*GAMs*) (Hastie & Tibshirani, 1987) form a class of statistical models that extends Generalized Linear Models (Nelder & Wedderburn, 1972) by incorporating non-linear transformations of each feature. Following Hastie & Tibshirani (1987), GAMs can be expressed as:

$$g\big(E(Y|X)\big) = \beta + \sum_{i=1}^{p} f_i(X_i), \qquad \text{(GAM)}$$

where $Y = (y_1, \ldots, y_N) \in \mathbb{R}^N$ is a vector of $N$ observed values of a target (random) variable, $X = [X_1, \ldots, X_p] \in \mathbb{R}^{N \times p}$ assembles the observed feature variables $X_i = (x_{i,1}, \ldots, x_{i,N}) \in \mathbb{R}^N$, and $f_i : \mathbb{R} \to \mathbb{R}$ are univariate, continuous *shape functions* modeling the individual feature transformations.[1] Furthermore, $\beta \in \mathbb{R}$ is a learnable global offset and $g : \mathbb{R} \to \mathbb{R}$ is the link function that relates the (expected) target value to the feature variables, e.g. the logit function in binary classification or the identity function in regression. The shape functions $f_i$ precisely describe the contribution of each individual feature variable in GAMs, and can be visualized and interpreted similarly to coefficients in a linear model. This allows practitioners to fully understand the learned prediction rule and gain further insights into the underlying data.

Whereas early GAMs primarily used splines (Hastie & Tibshirani, 1987) or boosted decision trees (Lou et al., 2012; 2013; Caruana et al., 2015) to model $f_i$, more recent GAMs such as *Neural Additive Models* (*NAMs*) (Agarwal et al., 2021) use multilayer perceptrons (MLPs) to fit the functions $f_i$, benefitting from the universal approximation capacity of neural networks (Cybenko, 1989) as well as the support of automatic differentiation frameworks (Paszke et al., 2019; Bradbury et al., 2018; Abadi et al., 2016) and hardware acceleration. As a result, one can now solve the GAM fitting problem

$$\min_{(f_1, \ldots, f_p) \in \mathcal{H}} \frac{1}{N} \sum_{l=1}^{N} L\big(Y, \beta + \sum_{i=1}^{p} f_i(X_i)\big)$$
(GAM-Fit)

by common deep learning optimization techniques such as mini-batch stochastic gradient descent (SGD). Here, $L : \mathbb{R} \times \mathbb{R} \to \mathbb{R}$ is a loss function and $\mathcal{H} \subset \{(f_1, \ldots, f_p) \mid f_i : \mathbb{R} \to \mathbb{R}\}$ any function class with differentiable parameters, e.g., MLPs or periodic functions like in NeuralProphet (Triebe et al., 2021).

### 2.2. Multicollinearity and Concurvity

Multicollinearity refers to a situation in which two or more feature variables within a linear statistical model are strongly correlated. Formally, this reads as follows:

**Definition 2.1** (Multicollinearity)**.** Let $X_1, \ldots, X_p \in \mathbb{R}^N$ be a set of feature variables where $X_i = (x_{i,1}, \ldots, x_{i,N}) \in \mathbb{R}^N$ represents $N$ observed values. We say that $X_1, \ldots, X_p$ are (perfectly) multicollinear if there exist $c_0, c_1, \ldots, c_p \in \mathbb{R}$, not all zero, such that $c_0 + \sum_{i=1}^{p} c_i X_i = 0$.

According to the above definition, every suitable linear combination of features can be modified by adding a trivial linear combination $c_0 + \sum_{i=1}^{p} c_i X_i = 0$. This can make individual effects of the features on a target variable difficult to disambiguate, impairing the interpretability of the fitted model. However, even in the absence of *perfect* multicollinearity, difficulties may arise.[2] For example, if two features are strongly correlated, estimating their individual contributions becomes challenging and highly sensitive to external noise. This typically results in inflated variance estimates for the linear regression coefficients (Ramsay et al., 2003), among other problems (Dormann et al., 2013).

The notion of concurvity was originally introduced in the context of GAMs to extend multicollinearity to non-linear feature transformations (Buja et al., 1989). In analogy with our definition of multicollinearity, we propose the following definition of concurvity:

**Definition 2.2** (Concurvity)**.** Let $X_1, \ldots, X_p \in \mathbb{R}^N$ be a set of feature variables and let $\mathcal{H} \subset \{(f_1, \ldots, f_p) \mid f_i : \mathbb{R} \to \mathbb{R}\}$ be a class of functions. We have (perfect) concurvity w.r.t. $X_1, \ldots, X_p$ and $\mathcal{H}$ if there exist $(g_1, \ldots, g_p) \in \mathcal{H}$ and $c_0 \in \mathbb{R}$ such that $c_0 + \sum_{i=1}^{p} g_i(X_i) = 0$ with $c_0, g_1(X_1), \ldots, g_N(X_N)$ not all zero.

Technically, concurvity simply amounts to the collinearity of the transformed feature variables, and Definition 2.1 is recovered when $\mathcal{H}$ is restricted to affine linear functions. Concurvity poses analogous challenges to multicollinearity: *Any* non-trivial zero-combination of features can be added to a solution of (GAM-Fit), rendering the fitted model less interpretable as each feature's contribution to the target is not immediately apparent. For further technical remarks on concurvity, we refer to Appendix A.2.

## 3. Concurvity Regularizer

Concurvity easily arises in relatively flexible GAMs, such as NAMs, since the mutual relationships between the functions $f_i$ are not constrained while fitting (GAM-Fit). This results in a large, degenerate search space with possibly infinitely many equivalent solutions. To remedy this problem, it appears natural to constrain the function space $\mathcal{H}$ of (GAM-Fit) such that the shape functions $f_i$ do not ex-

---

[1]As usual, when $f_i$ or $g$ are applied to a vector, their effect is understood elementwise.

[2]Informally, non-perfect multicollinearity describes situations where $\sum_{i=1}^{p} c_i X_i \approx 0$. However, a formal definition would also require an appropriate correlation or distance metric.

hibit spurious mutual dependencies. Here, our key insight is that *pairwise uncorrelatedness is sufficient to rule out concurvity*. Indeed, using $\mathcal{H}$ from Definition 2.2, let us consider the subclass

$$\mathcal{H}_\perp := \big\{ (f_1, \ldots, f_p) \in \mathcal{H} \mid \mathrm{Corr}\big(f_i(X_i), f_j(X_j)\big) = 0,$$
$$\forall i \neq j \big\} \subset \mathcal{H},$$

where $\mathrm{Corr}(\cdot, \cdot)$ is the Pearson correlation coefficient. It is not hard to see that concurvity w.r.t. $X_1, \ldots, X_p$ and $\mathcal{H}_\perp$ is impossible, regardless of the choice of $\mathcal{H}$ (for a proof, see Appendix A.1). From a geometric perspective, $\mathcal{H}_\perp$ imposes an orthogonality constraint on the feature vectors. The absence of concurvity follows from the fact that an orthogonal system of vectors is also linearly independent. However, it is not immediately apparent how to efficiently constrain the optimization domain of (GAM-Fit) to $\mathcal{H}_\perp$. Therefore, we rephrase the above idea as an unconstrained optimization problem:

$$\min_{(f_1, \ldots, f_p) \in \mathcal{H}} \frac{1}{N} \sum_{l=1}^{N} L\big(Y, \beta + \textstyle\sum_{i=1}^{p} f_i(X_i)\big) +$$
$$\lambda \cdot R_\perp(\{f_i\}_i, \{X_i\}_i),$$
$$(\text{GAM-Fit}_\perp)$$

where $R_\perp : \mathcal{H} \times \mathbb{R}^{N \times p} \to [0, 1]$ denotes our proposed *concurvity regularizer*:

$$R_\perp\big(\{f_i\}_i, \{X_i\}_i\big) :=$$
$$\frac{1}{p(p-1)/2} \sum_{i=1}^{p} \sum_{j=i+1}^{p} \big| \mathrm{Corr}\big(f_i(X_i), f_j(X_j)\big) \big|.$$

Using the proposed regularizer, (GAM-Fit$_\perp$) simultaneously minimizes the loss function and the distance to the decorrelation space $\mathcal{H}_\perp$. In situations where high accuracy and elimination of concurvity cannot be achieved simultaneously, a trade-off between the two objectives occurs, with the regularization parameter $\lambda \geq 0$ determining the relative importance of each objective. An empirical evaluation of this objective trade-off is presented in the subsequent experimental section.

Since $R_\perp$ is differentiable almost everywhere, (GAM-Fit$_\perp$) can be optimized with gradient descent and automatic differentiation. Additional computational costs arise from the quadratic scaling of $R_\perp$ in the number of additive components, although this can be efficiently addressed by parallelization. A notable difference to traditional regularizers like $\ell_1$- or $\ell_2$-penalties is the dependency of $R_\perp$ on the data $\{X_i\}_i$. As a consequence, the regularizer is also affected by the batch size and hence becomes more accurate with larger batches. An additional remark on our regularizer can be found in Appendix A.2.

Our concurvity regularizer is agnostic to the function class of $f_i$, hence the mostly spline-based concurvity metrics

proposed in the literature (Wood, 2001; Kovács, 2022) are not directly applicable. Similarly to Ramsay et al. (2003), we decide to report the average $R_\perp(\{f_i\}_i, \{X_i\}_i) \in [0, 1]$ as our metric of concurvity.

## 4. Experimental Evaluation

In order to investigate the effectiveness of our proposed regularizer, we will conduct evaluations using both synthetic and real-world datasets, with a particular focus on the ubiquitous applications of GAMs: tabular and time-series data. For the experiments involving synthetic and tabular data, we have chosen to use Neural Additive Models (NAMs), as they are differentiable and hence amenable to our regularizer. For time series data, we investigate NeuralProphet models which contain an additive component modeling seasonality. Further elaboration on our experimental setup, including detailed specifications and parameters, can be found in Appendix C.

### 4.1. Toy Examples

In the following, we design and investigate two instructive toy examples to facilitate a deeper comprehension of the proposed regularizer as well as the relationship between the regularization strength $\lambda$ and the corresponding model accuracy.

**Toy Example 1: Concurvity regularization with and without multicollinearity**  To compare the influence of concurvity regularization on model training in the presence of multicollinearity, we generate synthetic data according to the linear model $Y = 1 \cdot X_1 + 0 \cdot X_2$. We focus on two settings where the features $X_1$ and $X_2$ are either independently sampled from a uniform distribution (stochastically independent case) or fixed to identical samples (perfectly correlated case).

We first investigate the influence of concurvity regularization on the contribution of each feature to the target by measuring the correlation of the transformed features $f_1(X_1)$, $f_2(X_2)$ with the target variable $Y$. The results are shown in Figure 2b. In the stochastically independent case, the NAM accurately captures the relationship between input features and the target variable regardless of the regularization setting, as observed by the high correlation for $f_1(X_1)$ and zero correlation for $f_2(X_2)$ with the target. This result emphasizes the minimal impact of the regularizer when features are uncorrelated (c.f. Appendix A.2 for details). In the perfectly correlated case, the NAM trained without concurvity regularization displays a high correlation for both $f_1(X_1)$ and $f_2(X_2)$ with the target, thus using both features for its predictions. In contrast, when concurvity regularization is applied, the NAM is pushed towards approximately orthogonal $f_i$, which encourages feature selection, as indicated by

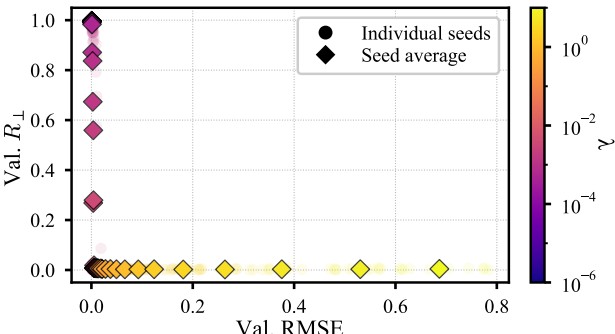

(a) Trade-Off Curve between model accuracy (validation RMSE) and measured concurvity (validation $R_\perp$). Results are averaged over 10 random initializations per regularization strength $\lambda$.

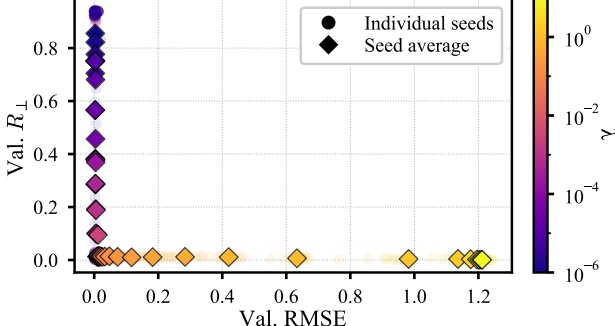

(a) Trade-Off Curve between model accuracy (validation RMSE) and measured concurvity ($R_\perp$).

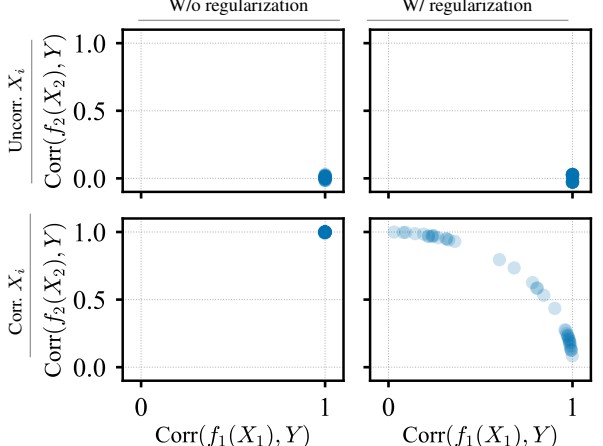

(b) Effect of concurvity regularization on uncorrelated and correlated features. For each of the four settings, 40 random initializations were evaluated.

*Figure 2.* Results for Toy Example 1.

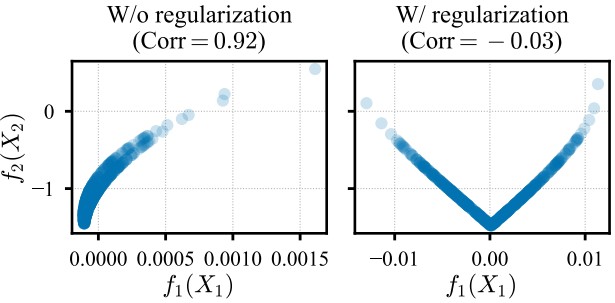

(b) Comparison of transformed feature correlation with and without concurvity regularization.

*Figure 3.* Results for Toy Example 2.

high correlation of either $f_1(X_1)$ or $f_2(X_2)$ with the target as there is no natural preference. This nicely illustrates the impact of the proposed regularization on correlated feature contributions. We further investigate the decorrelation under perfectly correlated features in Appendix D.1.

Secondly, and perhaps most importantly, we examine the trade-off between the validation RMSE and concurvity $R_\perp$ in Figure 2a. Our findings suggest that with moderate regularization strengths $\lambda$, we can effectively eradicate concurvity without compromising the accuracy of the model. It is only when the regularization strength is considerably high that the RMSE is adversely affected, without any further reduction in the measured concurvity.

**Toy Example 2: Concurvity regularization in the case of non-linearly dependent features**   Next, we evaluate our regularizer in the case of non-linear relationships between features, a setting to which it is equally applicable. To this

end, we design an experiment with feature variables that are uncorrelated, but not stochastically independent due to a non-linear relationship. We choose $X_1 = Z$ and $X_2 = |Z|$ where $Z$ is a standard Gaussian, and let $Y = X_2$ be the target variable. In this case, there is no multicollinearity by design, but the model may still learn perfectly correlated feature transformations. For example, a NAM could learn $f_1 = |\cdot|$ and $f_2 = \text{id}$, then $f_1(X_1) = f_2(X_2)$, which are fully correlated, yet provide a perfect fit. For our experiment, we use the same NAM model configuration as in the previous toy example.

The transformed features for the NAM fitted with and without regularization are visualized in Figure 3b. We find that the regularizer has effectively led the NAM to learn decorrelated feature transformations $f_1$ and $f_2$, reflected in the severe difference in feature correlation ($\text{Corr} = -0.03$ for the regularized NAM compared to $\text{Corr} = 0.92$ for the unregularized model). Moreover, these results suggest that the regularized model seems to have learned the relationship between the features, where $f_2(X_2)$ seems to approximate $|f_1(X_1)|$.

Finally, a trade-off curve of the validation RMSE and $R_\perp$ is shown in Figure 3a, illustrating that even in the case of

non-linearly dependent features, our proposed regularizer effectively mitigates the measured concurvity $R_\perp$ with minimal impact on the model's accuracy as measured by the RMSE.

### 4.2. Tabular Data

In our final series of experiments, we investigate the benefits of the proposed regularizer when applied to NAMs trained on real-world tabular datasets – a data type ubiquitous in the healthcare domain and often tackled with conventional machine learning methods such as random forests or gradient boosting. We concentrate our analysis on four well-studied datasets: MIMIC-II (Lee et al., 2011), MIMIC-III (Johnson et al., 2016), California Housing (Pace & Barry, 1997) and Adult (Dua & Graff, 2017). The datasets were selected with the aim of covering different dataset sizes as well as target variables (regression for California Housing and binary classification for the rest). NAMs are used throughout the evaluation, subject to a distinct hyperparameter optimization for each dataset. Details are presented in Appendix B, and additional results for Boston Housing (Harrison Jr & Rubinfeld, 1978) and Support2 in are provided in Appendix D.3.

First, we explore the trade-off between the concurvity measure $R_\perp$ and validation fit quality when employing the proposed concurvity regularization. Figure 4 displays the results for the tabular datasets. It is clear that the concurvity regularizer effectively reduces the concurvity measure $R_\perp$ without significantly compromising the model fit quality across all considered datasets, in particular in case of small to moderate regularization strengths. For example, on the California Housing dataset, we are able to reduce $R_\perp$ by almost an order of magnitude from around 0.2 to 0.05, while increasing the validation RMSE by about 10% from 0.59 to 0.66. Additionally, we observe the variation in the scale of $R_\perp$ across the datasets, exemplified by the MIMIC-III dataset where the transformed features show low correlation without regularization, as well as a rather small reduction in concurvity caused by regularization. In practice, trade-off curves between concurvity and model accuracy can serve as a valuable tool for identifying the optimal level of regularization strength, e.g. via the elbow technique (Thorndike, 1953).

**Case study: MIMIC-II** Our preceding experiments demonstrated that concurvity reduction can be achieved when training NAMs on tabular data. However, the practical significance of this observation in relation to interpretability remains unclear. To address this, we perform a more detailed analysis of NAMs trained on the MIMIC-II dataset, a publicly available critical care dataset used for predicting mortality risk from a number of demographic and biophysical indcators. In the following analysis, we compare NAMs trained with and without concurvity regularization. More

specifically, we evaluate $\lambda = 1.0$ (determined based on Figure 4) and $\lambda = 0.0$ both for 64 random weight initializations.

First, we assess the effect of the regularizer on the model fit, finding that regularization increases the mean test ROC AUC by about 5% from about 0.80 to 0.84 and slightly increases the spread between the seeds, as shown in Figure 6b. Note that the result in the non-regularized case is on par with the original NAM evaluation (Chang et al., 2022) serving as a sanity check of our experimental setup.

Second, we juxtapose the feature correlations of non-linearly transformed features for models trained with and without regularization. The results, as displayed in Figure 5a (upper right triangular matrices), are contrasted with the raw input feature correlations (lower left triangular matrices). It is evident that without regularization, high input correlations tend to result in correlated transformed features, as seen in the left correlation matrix. Conversely, the right correlation matrix reveals that concurvity regularization effectively reduces the correlation of transformed features. This effect is especially pronounced for previously pairwise correlated features such as *Urea*, *Renal* and *K*.

Third, we investigate how concurvity impacts the estimation of the individual feature importances, which is of key interest for interpretable models such as NAMs. Following Agarwal et al. (2021), we measure the importance of feature $i$ as $\frac{1}{N} \sum_{j=1}^{N} |f_i(x_{ij}) - \overline{f_i}|$ where $\overline{f_i}$ denotes the average of shape function $f_i$ over the training datapoints. We visualize the distribution of feature importances over our regularized and unregularized ensembles of NAMs in Figure 5b. Interestingly, regularization seems to enforce sparsity in the feature importances, pushing several features towards zero importance while a few stronger features grow in importance. Though sparsity is not necessarily an effect of concurvity regularization, decorrelation can lead to feature selection in the case of mutual dependencies between features. See Appendix D.3.1 for an analogous analysis of the California Housing dataset, showing partially different effects of regularization on feature importances.

With regards to the varying effect of regularization on the respective features, two observations are particularly interesting: (1) Features that are mostly uncorrelated remain unaffected by regularization – an effect we have previously seen in Toy Example 1 – which can, for example, be observed in the case of the *Metastatic Cancer* feature. (2) Input correlations in this case lead to a clear feature selection: the importances of *Urea* and *K* are effectively pruned from the model whereas *Renal*, with which they both correlate, gains in importance with regularization. Similarly to Toy Example 1, we see that the regularizer encourages feature selection for correlated feature pairs.

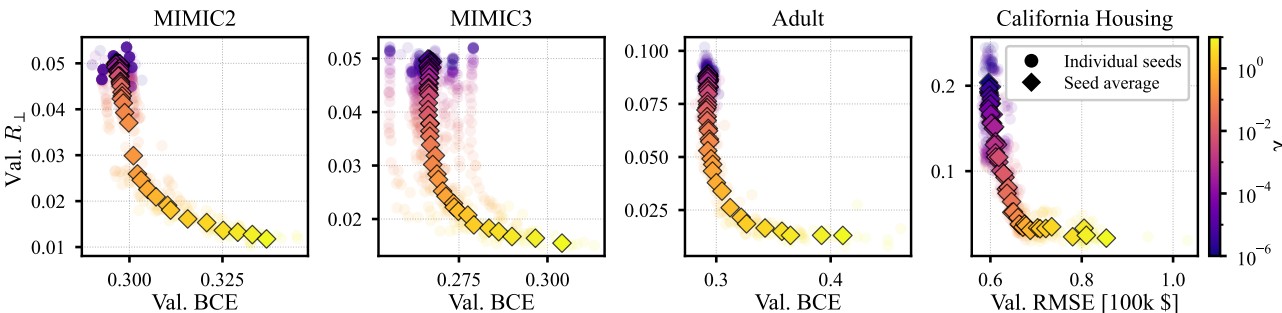

*Figure 4.* Trade-off curves between model fit quality and measured concurvity $R_\perp$ for 50 levels of concurvity regularization strength $\lambda$. Each regularization strength is evaluated over 10 initialization seeds to account for training variability.

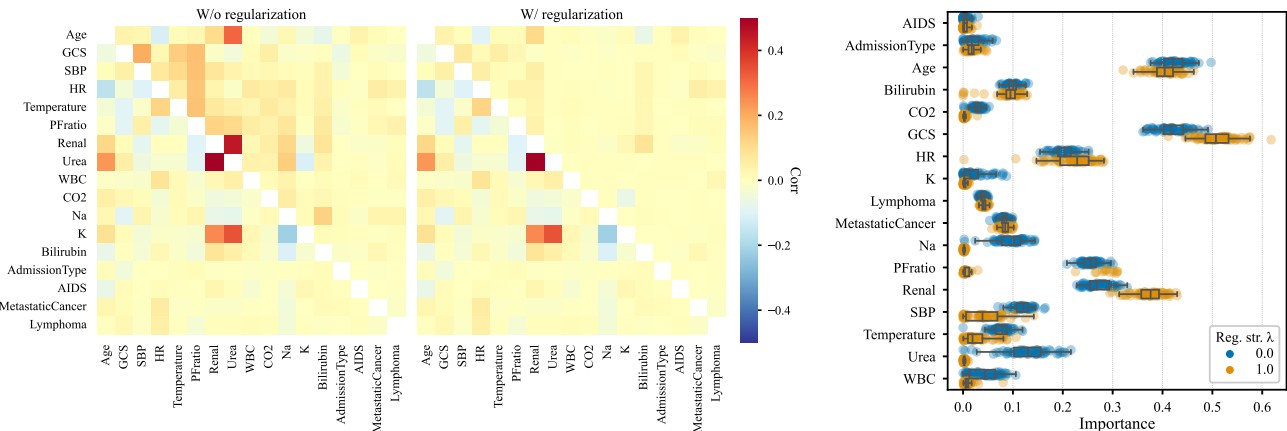

(a) Average feature correlations of the features (lower left) and non-linearly transformed features (upper right).

(b) Combined box and strip plot of the models' feature importances.

*Figure 5.* Results for the MIMIC-II dataset.

Finally, to visualize the impact of the regularization on model interpretability in more detail, the shape functions of three features are shown in Figure 6. Here, the features *K* and *Urea* are positively correlated in the input space but negatively correlated in the output space (c.f. Figure 5a), indicating possible self-canceling behaviour. This problem is effectively mitigated by the proposed regularization, removing opposing contributions. Instead, we see an increased effect of *Renal*, correlated with both features in the input space. For comparison, the contribution of the less strongly correlated *Age* feature remains virtually unchanged by the regularization. Similar behavior can be observed for the remaining feature contributions, which are depicted in Appendix D.3.

In summary, our case study on MIMIC-II illustrates that concurvity regularization produces more compact representations in a NAM in terms of shape functions and feature importance, whilst maintaining high model accuracy. Based on the data alone we can only obtain correlational results and potential causal links between the features and target may require the evaluation of a domain expert, i.e., a doctor,

which is outside the scope of the current work. However, concurvity regularization ensures that we are not putting flawed questions to the experts because of misleading correlations in the transformed features.

## 5. Related Work

**Classical works on concurvity in GAMs** The term concurvity was first introduced by Buja et al. (1989); for a well-written introduction to concurvity we refer the reader to Ramsay et al. (2003). Numerous subsequent works have developed techniques to address concurvity, such as improving numerical stability in spline-based GAM fitting (Wood, 2008; He, 2004) and adapting Lasso regularization for GAMs (Avalos et al., 2003). Partial GAMs (Gu et al., 2010) were proposed to address concurvity through sequential maximization of Mutual Information between response variables and covariates. More recently, Kovács (2022) compared several feature selection algorithms for GAMs and found algorithms selecting a larger feature set to be more susceptible to concurvity, a property first noticed by Hall

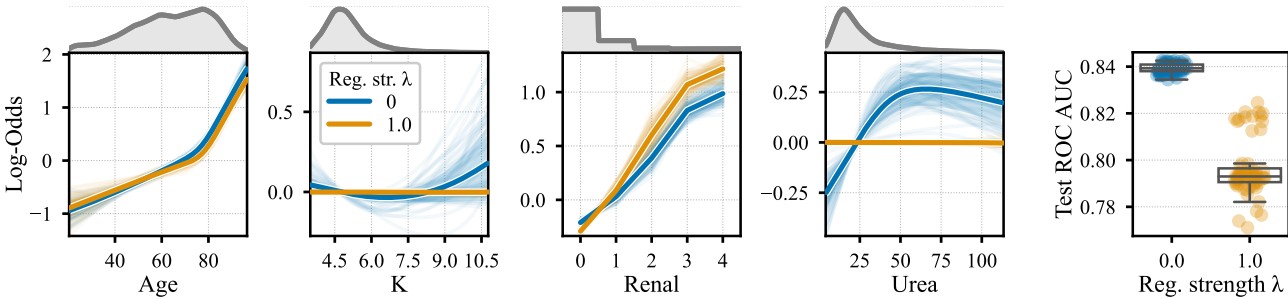

(a) Average and individual shape functions of 3 out of the 9 selected features. A kernel density estimate of the training data distribution is depicted on top.

(b) Test ROC AUC.

*Figure 6.* Results for the MIMIC-II dataset.

(1999). In addition, Kovács (2022) proposes a novel feature selection algorithm that chooses a minimal subset to deal with concurvity. In contrast, our proposed regularizer adds no additional constraints on the feature set size and does not specifically enforce feature sparsity. We refer to Kovács (2022) for a comparison of different concurvity metrics.

**Modern neural approaches to GAMs** Recent advancements in neural approaches to GAMs, such as NAMs (Agarwal et al., 2021) and NeuralProphet (Triebe et al., 2021), have provided more flexible and powerful alternatives to classical methods (Hastie & Tibshirani, 1987). These have spurred interest in the subject leading to several extensions of NAMs (Chang et al., 2022; Dubey et al., 2022; Radenovic et al., 2022; Xu et al., 2022; Enouen & Liu, 2022). Our approach is compatible with the existing methodologies, and can be readily integrated if they are implemented in an automatic differentiation framework.

**Regularization via decorrelation** Similar types of decorrelation regularizers have previously been proposed in the machine learning literature but in different contexts. Cogswell et al. (2016) found that regularizing the cross-covariance of hidden activations significantly increases generalization performance and proposed DeCov. OrthoReg (Rodríguez et al., 2017) was proposed to regularize negatively correlated features to increase generalization performance by reducing redundancy in the network. Similarly, Xie et al. (2017) propose to add a regularizer enforcing orthonormal columns in weight matrices. More recent approaches, such as Barlow Twins (Zbontar et al., 2021), utilize decorrelation as a self-supervised learning technique to learn representations that are invariant to different transformations of the input data.

## 6. Conclusion

In this paper, we have introduced a differentiable concurvity regularizer, designed to mitigate the often overlooked issue of concurvity in differentiable Generalized Additive Models (GAMs). Through comprehensive empirical evaluations, we demonstrated that our regularizer effectively reduces concurvity in differentiable GAMs such as Neural Additive Models and NeuralProphet. This in turn significantly enhances the interpretability and reliability of the learned feature functions, a vital attribute in various safety-critical and strongly regulated applications. Importantly, our regularizer achieves these improvements while maintaining high prediction quality, provided it is carefully applied. We underscore that while the interpretability-accuracy trade-off is an inherent aspect of concurvity regularization, the benefits of increased interpretability and consistent feature importances across model initializations are substantial, particularly in real-world decision-making scenarios.

An intriguing avenue for future work could be to examine the impact of our regularizer on fairness in GAMs. While prior work (Chang et al., 2021) suggests that GAMs with high feature sparsity can miss patterns in the data and be unfair to minorities, our concurvity regularizer does not directly enforce feature sparsity. Thus, a comparison between sparsity regularizers and our concurvity regularizer in unbalanced datasets would be of high interest. In addition, future work could explore how the joint optimization of concurvity and model fit could be improved by framing it as a multi-objective problem. Moreover, it would be interesting to see how the concurvity regularizer works in NAMs that incorporate pairwise interactions. Specifically, contrasting this with the ANOVA decomposition proposed by Lengerich et al. (2020), in terms of single and pairwise interactions, could unveil some interesting results.

We conclude by encouraging researchers and practitioners to "curve your enthusiasm" – that is, to seriously consider concurvity in GAM modeling workflows. We believe this will lead to more interpretable models and hence more reliable and robust analyses, potentially avoiding false conclusions.

## 7. Acknowledgements

We would like to thank Aaron Klein, Roland Zimmermann, Felix Pieper, Ziyad Sheebaelhamd, Thomas Wollmann, Fabio Haenel, Sebastian Schulze and Christian Leibig for their constructive feedback in preparation of this paper.

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

# A. Additional Remarks and Theoretical Results for the Proposed Regularizer

## A.1. The Decorrelation Space $\mathcal{H}_\perp$ Rules Out Concurvity

In this section, we formalize our claim from Section 3 that the space $\mathcal{H}_\perp$ provably rules out concurvity. In the following, $\langle \cdot, \cdot \rangle$ and $\| \cdot \|_2$ denote the Euclidean scalar product and norm, respectively. For $v = (v_1, \ldots, v_N) \in \mathbb{R}^N$, we set $\bar{v} := \frac{1}{N} \sum_{l=1}^{N} v_l$ and denote the all-one vector by $\mathbb{1} := (1, \ldots, 1) \in \mathbb{R}^N$. The standard Pearson's correlation coefficient is then given by[3]

$$\mathrm{Corr}(v, w) := \begin{cases} \frac{\langle v - \bar{v}\mathbb{1}, w - \bar{w}\mathbb{1} \rangle}{\|v - \bar{v}\mathbb{1}\|_2 \cdot \|w - \bar{w}\mathbb{1}\|_2} & \text{if } v \text{ and } w \text{ are non-constant,} \\ \infty & \text{otherwise,} \end{cases} \quad v, w \in \mathbb{R}^N.$$

**Lemma A.1.** *Let $X_1, \ldots, X_p \in \mathbb{R}^N$ be a set of feature variables with $p > 1$ and let $\mathcal{H} \subset \{(f_1, \ldots, f_p) \mid f_i : \mathbb{R} \to \mathbb{R}\}$ be a class of functions. Consider the following subclass of $\mathcal{H}$:*

$$\mathcal{H}_\perp := \{(f_1, \ldots, f_p) \in \mathcal{H} \mid \mathrm{Corr}(f_i(X_i), f_j(X_j)) = 0 \text{ for all } i \neq j \} \subset \mathcal{H}.$$

*Then we do* not *have concurvity w.r.t. $X_1, \ldots, X_p$ and $\mathcal{H}_\perp$.*

*Proof.* Towards a contradiction, assume that there are $(g_1, \ldots, g_p) \in \mathcal{H}_\perp$ and $c_0 \in \mathbb{R}$ such that $c_0 \mathbb{1} + \sum_{i=1}^{p} g_i(X_i) = 0$.

We set $v_i := g_i(X_i)$ and trivially add the mean vectors to the linear combination:

$$0 = c_0 \mathbb{1} + \sum_{i=1}^{p} v_i = \left(c_0 + \sum_{i=1}^{p} \bar{v}_i\right) \mathbb{1} + \sum_{i=1}^{p} (v_i - \bar{v}_i \mathbb{1}).$$

Using that $\mathrm{Corr}(v_i, v_j) = 0$ for $i \neq j$, we then obtain

$$0 = \langle 0, v_j - \bar{v}_j \mathbb{1} \rangle = \left(c_0 + \sum_{i=1}^{p} \bar{v}_i\right) \langle \mathbb{1}, v_j - \bar{v}_j \mathbb{1} \rangle + \sum_{i=1}^{p} \langle v_i - \bar{v}_i \mathbb{1}, v_j - \bar{v}_j \mathbb{1} \rangle$$

$$= \left(c_0 + \sum_{i=1}^{p} \bar{v}_i\right) \cdot (N\bar{v}_j - N\bar{v}_j) + \|v_j - \bar{v}_j \mathbb{1}\|_2^2 = \|v_j - \bar{v}_j \mathbb{1}\|_2^2.$$

We conclude that $v_j = \bar{v}_j \mathbb{1}$, i.e., $v_j$ is a constant vector. But this contradicts the definition of $\mathcal{H}_\perp$ because we would have $\mathrm{Corr}(v_i, v_j) = \infty$ with any $i \neq j$. $\square$

## A.2. Additional Remarks on Concurvity and our Regularizer

(1) The definitions of multicollinearity (Definition 2.1) and concurvity (Definition 2.2) are based on a fixed (deterministic) feature design, but one could also formulate probabilistic versions, cf. (Signoretto et al., 2008). The latter typically facilitates a theoretical analysis, which, however, is not the focus of our work. Moreover, a probabilistic definition would not cover an important practical source of multicollinearity, namely underdetermined systems where $N < K$.

(2) Although closely related, multicollinearity does not necessarily imply concurvity and vice versa. Indeed, one can easily come up with setups where perfectly correlated features become decorrelated after a non-linear transform. Similarly, uncorrelated input features can be made perfectly correlated with a non-linearity. Two simple (toy) examples are presented in Section 4.1.

(3) Our concurvity regularizer $R_\perp$ does not automatically affect the predictive performance of a GAM. For example, assuming that the input features are drawn from stochastically independent random variables, we can conclude that $\mathrm{Corr}(f_i(X_i), f_j(X_j)) \approx 0$ for a large enough sample size $N$, since non-linear transforms of independent random variables remain independent. Consequently, we have that $R_\perp(\{f_i\}_i, \{X_i\}_i) \approx 0$, so that no (in this case undesirable) regularization takes effect.

---

[3]The special case of constant vectors could be treated differently, e.g., by setting the correlation to $0$. The version we use here is most convenient for our purposes as it excludes the treatment of additional special cases in Lemma A.1.

| Hyperparameter | Value / Range | Scaling |
|---|---|---|
| Learning Rate | [1e-4, 1e-1] | log |
| Weight Decay | [1e-6, 1] | log |
| Activation | [ELU, GELU, ReLU] | cat. |
| # of neurons per layer | [2, 256] | linear |
| # of hidden layers | [1, 6] | linear |
| Num. Epochs | [10, 500] | linear |

*Table 1.* Hyperparameter Search Space

| Hyperparameter | Value |
|---|---|
| Learning Rate | 1e-3 |
| Weight Decay | 0.0 |
| Activation | GELU |
| # of neurons per layer | 128 |
| # of hidden layers | 3 |
| Num. Epochs | 50 |
| Batch Size | 128 |
| Correlation Denominator Eps | 1e-12 |
| Start Conc. Reg. after x% of steps | 5 |

(a) Toy Example 1&2

| Hyperparameter | Value |
|---|---|
| Learning Rate | 7.93e-4 |
| Weight Decay | 1.79e-2 |
| Activation | ELU |
| # of neurons per layer | 75 |
| # of hidden layers | 6 |
| Num. Epochs | 91 |
| Batch Size | 128 |
| Correlation Denominator Eps | 1e-12 |
| Start Conc. Reg. after x% of steps | 5 |

(b) Boston Housing

| Hyperparameter | Value |
|---|---|
| Learning Rate | 9.46e-3 |
| Weight Decay | 3.73e-3 |
| Activation | ReLU |
| # of neurons per layer | 72 |
| # of hidden layers | 5 |
| Num. Epochs | 39 |
| Batch Size | 512 |
| Correlation Denominator Eps | 1e-12 |
| Start Conc. Reg. after x% of steps | 5 |

(c) California Housing

| Hyperparameter | Value |
|---|---|
| Learning Rate | 2.64e-3 |
| Weight Decay | 1.64e-3 |
| Activation | GELU |
| # of neurons per layer | 204 |
| # of hidden layers | 4 |
| Num. Epochs | 200 |
| Batch Size | 512 |
| Correlation Denominator Eps | 1e-12 |
| Start Conc. Reg. after x% of steps | 5 |

(d) Adult

| Hyperparameter | Value |
|---|---|
| Learning Rate | 3.31e-3 |
| Weight Decay | 1.08e-3 |
| Activation | GELU |
| # of neurons per layer | 190 |
| # of hidden layers | 3 |
| Num. Epochs | 20 |
| Batch Size | 512 |
| Correlation Denominator Eps | 1e-12 |
| Start Conc. Reg. after x% of steps | 5 |

(e) MIMIC-II

| Hyperparameter | Value |
|---|---|
| Learning Rate | 3.88e-3 |
| Weight Decay | 1.10e-3 |
| Activation | GELU |
| # of neurons per layer | 168 |
| # of hidden layers | 3 |
| Num. Epochs | 23 |
| Batch Size | 512 |
| Correlation Denominator Eps | 1e-12 |
| Start Conc. Reg. after x% of steps | 5 |

(f) MIMIC-III

| Hyperparameter | Value |
|---|---|
| Learning Rate | 3.42e-3 |
| Weight Decay | 1.05e-2 |
| Activation | GELU |
| # of neurons per layer | 127 |
| # of hidden layers | 3 |
| Num. Epochs | 30 |
| Batch Size | 512 |
| Correlation Denominator Eps | 1e-12 |
| Start Conc. Reg. after x% of steps | 5 |

(g) Support2

*Figure 7.* Hyperparameters per dataset for NAM experiments.

## B. Hyperparameter Optimization

### B.1. Tabular Datasets

For hyperparameter optimization we use Tree-structured Parzen Estimator (TPE) (Bergstra et al., 2011) as implemented in Optuna (Akiba et al., 2019). We run the optimization for a budget of 500 function evaluations and optimize w.r.t. validation RMSE for Boston Housing and California Housing or validation binary cross entropy loss for Adult.

The hyperparameter space and default parameters are shown in Table 1 and the hyperparameters per dataset are shown in Figure 7.

## C. Experimental Details

### C.1. NAM experiments

In all of our NAM experiments, we use the AdamW optimizer (Loshchilov & Hutter, 2019) and adjust our learning rate using Cosine Annealing (Loshchilov & Hutter, 2017) and decay to 0. For all regression problems we use the Mean Squared Error (MSE) and for all binary classfication problems the Binary Cross Entropy (BCE) as loss function $L$.

In all of our experiments, we add the concurvity regularization only after 5% of the total optimization steps for stability reasons.

## C.2. Toy Examples

We sample 10000 datapoints from the model and use 7000, 2000, 1000 for training, validation and testing respectively. We use the validation split to find adequate hyperparameters via a small manual search. Our NAM has 3 layers per feature NN with 128 hidden units and uses the GeLU (Hendrycks & Gimpel, 2016) activation function. We use no weight decay, train for 50 epochs at an initial learning rate of 1e-3 and a batch size of 128. Otherwise the experimental setup is the same as described earlier in Section C.1.

# D. Additional results

## D.1. Toy Example 1

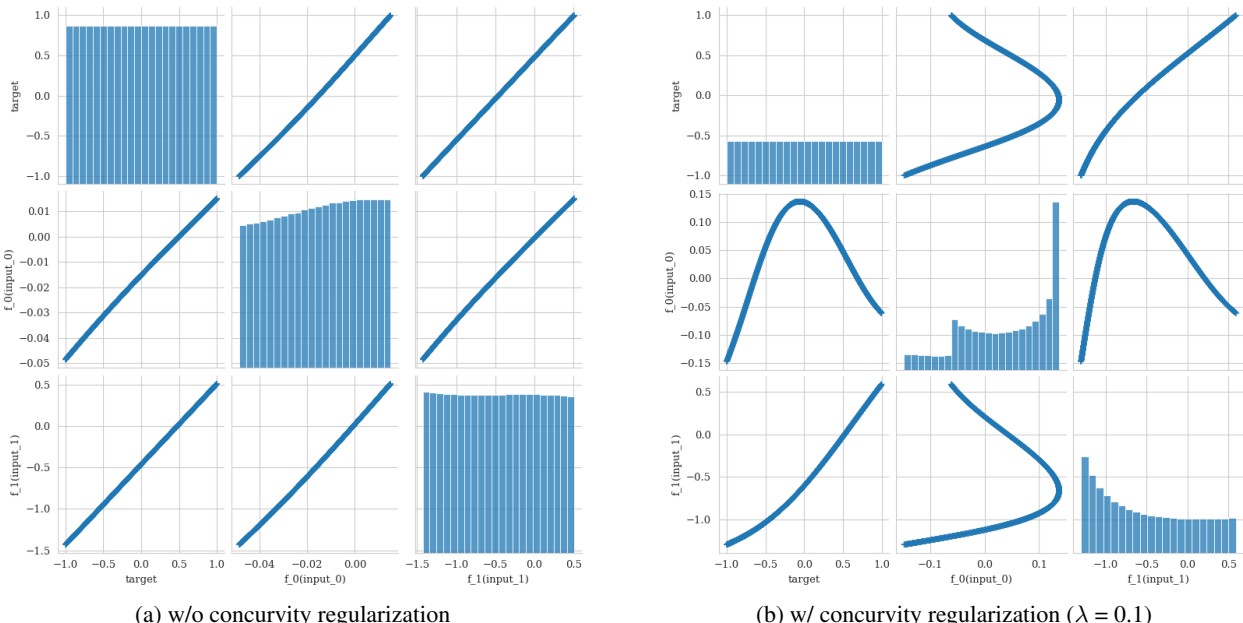

(a) w/o concurvity regularization          (b) w/ concurvity regularization ($\lambda = 0.1$)

*Figure 8.* (Toy Example 1) Pair plot demonstrating the difference in shape functions learned without and with concurvity regularization. The features $x_1$ and $x_2$ are fully correlated in this example.

In figure 8, we further visualize the decorrelation under perfectly correlated features for Toy Example 1. The figure shows an additional scatter plot comparing contributions of the transformed features.

## D.2. Time-Series Data

In this section, we provide context and additional results for the motivational example in Figure 1 on time-series forecasting using NeuralProphet (Triebe et al., 2021) which decomposes a time-series into various additive components such as seasonality or trend. In NeuralProphet, each seasonality $S_p$ is modeled using periodic functions as

$$S_p(t) = \sum_{j=1}^{k} a_j \cos\left(2\pi jt/p\right) + b_j \sin\left(2\pi jt/p\right)$$

where $k$ denotes the number of Fourier terms, $p$ the periodicity, and $a_j$, $b_j$ the trainable parameters of the model. In the motivational example depicted in Figure 1, we restrict the NeuralProphet model to two components, namely a weekly and daily seasonality. Our overall reduced NeuralProphet model is hence given by:

$$\hat{y}_t = S_{24h}(t) + S_{7d}(t).$$

If $k$ is sufficiently large, it can cause the frequency ranges of $S_{24h}$ and $S_{7d}$ to overlap, leading to concurvity in the model. The default values of $k$ provided by NeuralProphet for $S_{24h}$ ($k = 6$) and $S_{7d}$ ($k = 3$) are intentionally kept low to avert such

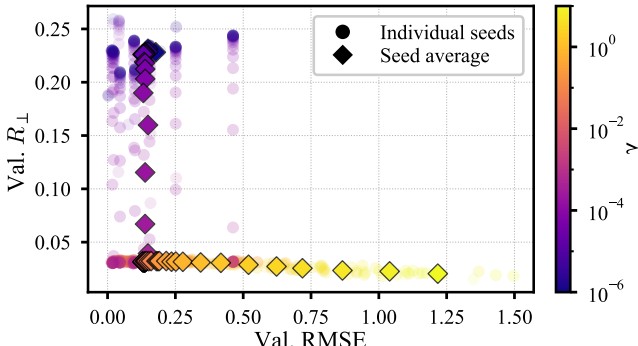

*Figure 9.* Trade-off curve for NeuralProphet model trained on step-function data.

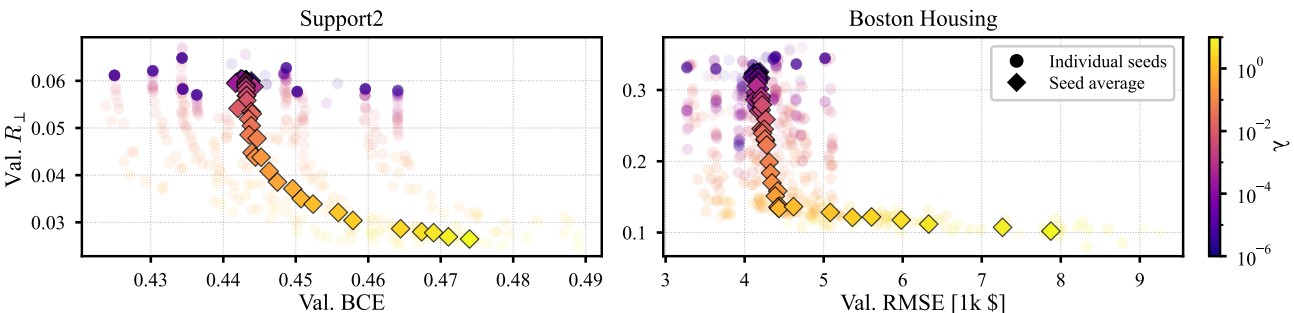

*Figure 10.* Tradeoff curves for Support2 and Boston Housing.

frequency overlap, as demonstrated in Figure 1 (left). However, this safety measure comes at the cost of model accuracy due to reduced model complexity.

Analogously to the previous examples, we present a trade-off curve between RMSE and concurvity, averaging over 10 random initialization seeds per regularization strength $\lambda$. In this experiment, we choose $k = 400$ components for both daily and weekly seasonality, to allow concurvity to occur and fit the data almost exactly. Our findings are identical to the toy examples, demonstrating a steep decline in concurvity when increasing $\lambda$ with only a small increase in RMSE.

Finally, we note that concurvity can often be identified by visual inspection for additive univariate time-series models as each component is a function of the same variable, c.f. Figure 1. In contrast, on multivariate tabular data, concurvity may go unnoticed if left unmeasured and hence lead to false conclusions, as we investigate next.

### D.3. Tabular Data

Additional trade-off curves for Support2 and Boston Housing are depicted in Figure 10. We also show the remaining shape functions for MIMIC-II in figure 11 that were not included in the case study before.

#### D.3.1. CASE STUDY: CALIFORNIA HOUSING

This section contains a more detailed analysis of NAMs trained on the California Housing dataset. In the following analysis, we compare NAMs trained with and without concurvity regularization. More specifically, we evaluate $\lambda = 0.1$ (determined based on Figure 4) and $\lambda = 0.0$ both for 60 random weight initializations.

First, we assess the effect of the regularizer on the model fit, finding that regularization increases the mean test RMSE by about 10% from about 0.58 to 0.64 and slightly decreases the spread between the seeds, as shown in Figure 13b. Note that the result in the non-regularized case is on par with the original NAM evaluation (Agarwal et al., 2021) serving as a sanity check of our experimental setup.

Second, we juxtapose the feature correlations of non-linearly transformed features for models trained with and without

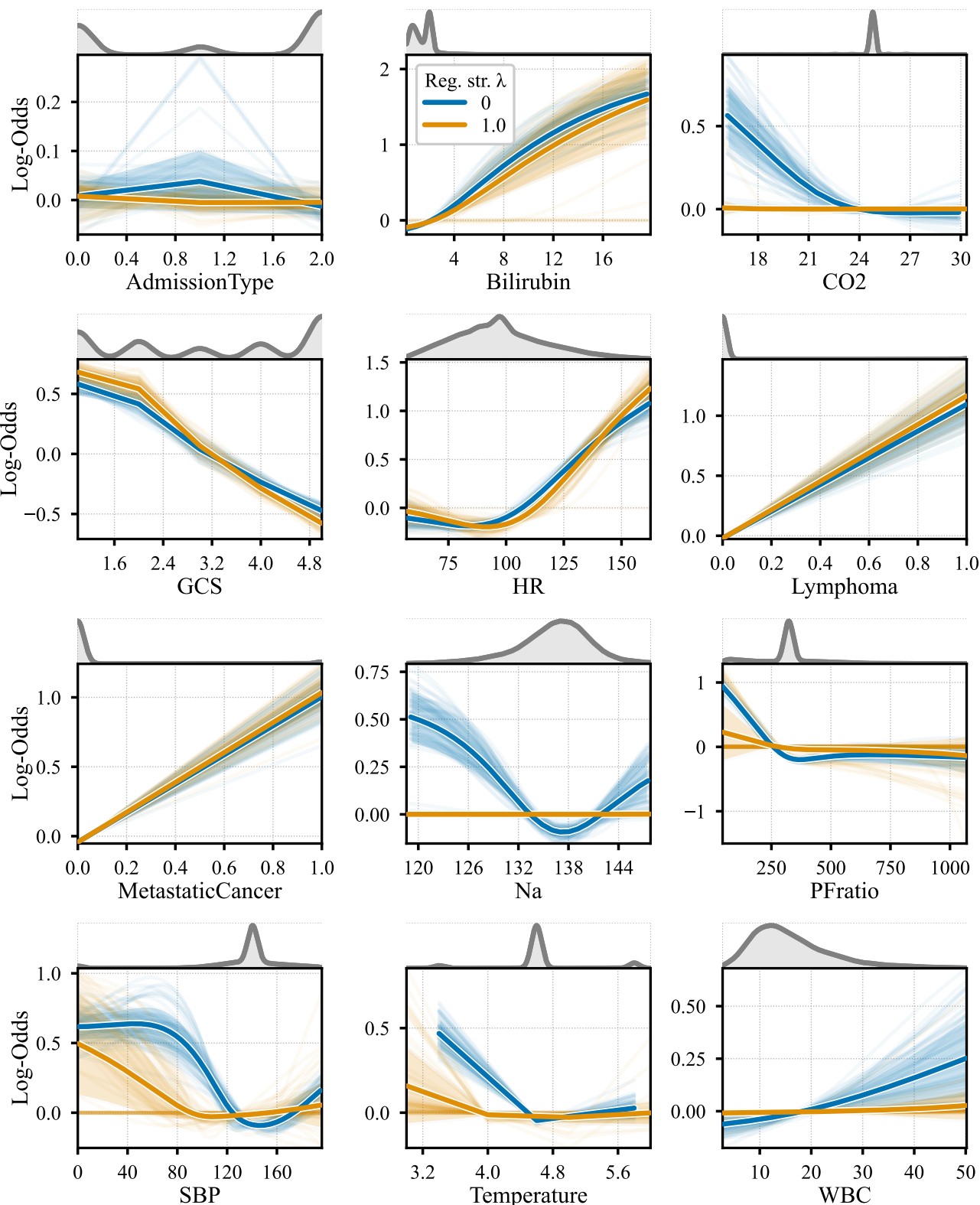

*Figure 11.* Additional average and individual shape functions for MIMIC-II. A kernel density estimate of the training data distribution is depicted on top.

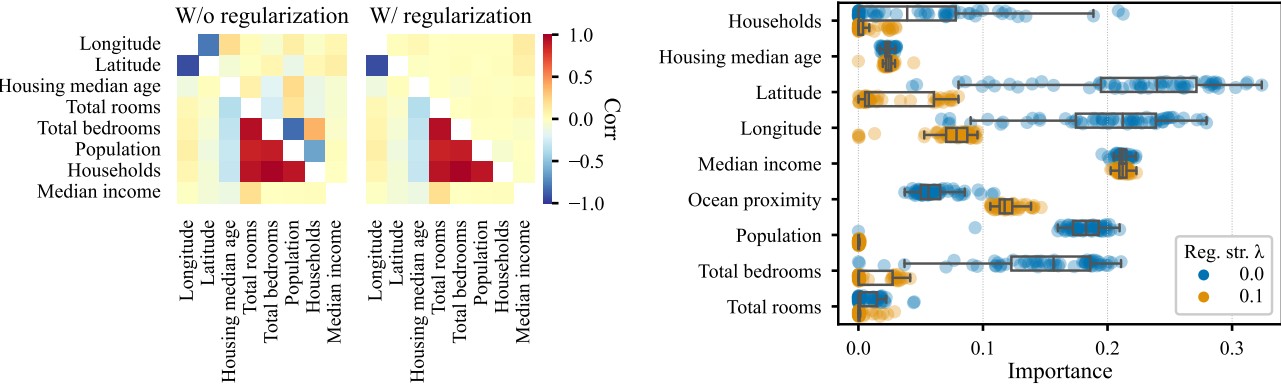

(a) Average feature correlations of the features (lower left) and non-linearly transformed features (upper right).

(b) Combined box and strip plot of the models' feature importances.

*Figure 12.* Results in the case of the California Housing dataset.

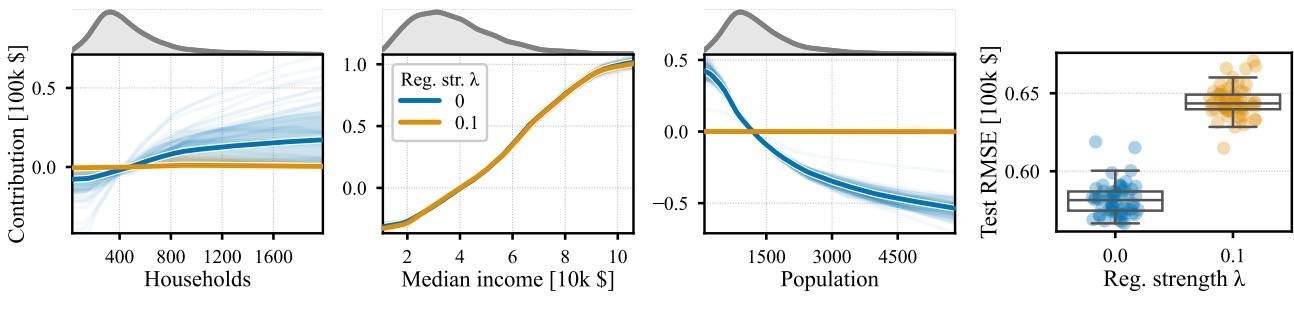

(a) Average and individual shape functions of 3 out of the 9 selected features. A kernel density estimate of the training data distribution is depicted on top.

(b) Test RMSEs.

*Figure 13.* Results in the case of the California Housing dataset. The considered NAMs were trained with and without concurvity regularization using 60 model initialization seeds each.

regularization. The results, as displayed in Figure 12a (upper right triangular matrices), are contrasted with the raw input feature correlations (lower left triangular matrices). It is evident that without regularization, high input correlations tend to result in correlated transformed features, as seen in the left correlation matrix. Conversely, the right correlation matrix reveals that concurvity regularization effectively reduces the correlation of transformed features. This effect is especially pronounced for previously highly correlated features such as *Longitude* and *Latitude*, or *Total Bedrooms* and *Households*.

Third, we investigate how concurvity impacts the estimation of the individual feature importances, which is of key interest for interpretable models such as NAMs. Following (Agarwal et al., 2021), we measure the importance of feature $i$ as $\frac{1}{N} \sum_{j=1}^{N} |f_i(x_{ij}) - \overline{f_i}|$ where $\overline{f_i}$ denotes the average of shape function $f_i$ over the training datapoints. We visualize the distribution of feature importances over our regularized and unregularized ensembles of NAMs in Figure 12b. It is apparent that feature importances tend to have a larger variance in the unregularized case compared to the regularized case, a pattern which is particularly clear for the strongly correlated features which we identified in Figure 12a. Such variance in feature importance can detrimentally impair the interpretability of the models, due to potential inconsistencies arising in absolute importance orders. However, our proposed concurvity regularizer effectively counteracts this issue, resulting in more consistent and compact feature importances across different random seeds. With regards to the varying effect of regularization on the respective features, two observations are particularly interesting: (1) Features that are mostly uncorrelated remain unaffected by the regularization – an effect we have previously seen in Toy Example 1 – which can, for example, be observed in the case of the *Median income* feature. (2) Input correlations lead to a bi-modal distribution in the corresponding feature importance as for example observable in the case of the *Longitude* and *Latitude* or *Total bedrooms* and *Households* features. Similarly to Toy Example 1, we see that the regularizer encourages feature selection for correlated feature pairs.

Finally, to visualize the impact of the regularization on model interpretability in more detail, the shape functions of three features are shown in Figure 13. Here, the features *Households* and *Population* are strongly negatively correlated (c.f. Figure 12a) which leads to their feature contributions largely canceling each other out. This problem is effectively mitigated by the proposed regularization, revealing naturally low contributions for both features. For comparison, the contribution of the mostly non-correlated *Median income* feature remains virtually unchanged by the regularization. Similar behavior can be observed for the remaining feature contributions, which are depicted in Appendix D.3.

In summary, our case study on the California Housing dataset establishes that concurvity regularization significantly enhances interpretability and consistency of a GAM in terms of shape functions and feature importance, whilst maintaining high model accuracy.

Additional results for California Housing are depicted in Figure 14.

## E. Dataset Details

**Boston Housing**   The Boston Housing Dataset (Harrison Jr & Rubinfeld, 1978), compiled by Harrison and Rubinfeld in the 1970s, is a benchmark dataset employed in machine learning and statistical modeling for housing price prediction. Consisting of 506 neighborhoods in the Boston metropolitan area, it features 13 attributes, including crime rate, zoning information, industrial acreage, Charles River proximity, air quality, housing characteristics, accessibility to employment centers, highways, and education, as well as demographic factors. The primary objective is to predict the median value of owner-occupied homes using these features.

**California Housing**   The California Housing Dataset (Pace & Barry, 1997) is a widely-used benchmark dataset for machine learning and statistical modeling, particularly in the domain of housing price prediction. Originally derived from the 1990 California Census, it consists of 20,640 samples, each representing a census block group. The dataset contains information on various housing-related attributes, such as median income, housing median age, average number of rooms, average number of bedrooms, population and average household size. It also includes the geographical location (latitude and longitude) of each block group. The objective is to predict the median house value. We obtained the dataset from Dua & Graff (2017).

**Adult**   The Adult Dataset (Dua & Graff, 2017) is also known as the "Census Income" dataset. Extracted from the 1994 United States Census Bureau data, it comprises 48,842 records, each representing an individual. The dataset contains 14 features, including age, work class, education, marital status, occupation, relationship, race, sex, capital gain, capital loss, hours worked per week, and native country. The objective is to predict whether an individual's annual income exceeds $50,000.

**MIMIC-II**   The MIMIC-II (Multiparameter Intelligent Monitoring in Intensive Care) dataset (Lee et al., 2011) is a public database, offering clinical data from a multitude of Intensive Care Unit (ICU) patients. It is managed by the MIT Lab for Computational Physiology and encompasses a wide variety of data points, such as patient demographics, vital signs, lab test results, medications, procedures, caregiver notes, and imaging reports, as well as mortality rates both in and out of the hospital.

**MIMIC-III**   MIMIC-III (Medical Information Mart for Intensive Care III) (Johnson et al., 2016) is the successor to the MIMIC-II database. It contains additional, more recent patient records and provides more detailed data, including free-text interpretation of imaging reports, allowing for more granular research and improved application in areas like machine learning, health informatics, and predictive modeling.

**SUPPORT 2**   The SUPPORT 2 (Study to Understand Prognoses and Preferences for Outcomes and Risks of Treatments) dataset is a clinical database that contains detailed medical information from a large cohort of seriously ill hospitalized adults. The dataset was created as part of a multi-center study designed to understand the outcomes of decisions made in the course of medical treatment. It provides extensive variables, including demographic data, physiological measurements, diagnostic information, treatment plans, and outcomes such as survival and quality of life. The data collected span a diverse set of medical conditions, making the SUPPORT 2 dataset a valuable resource for researchers seeking to study clinical decision-making, prognosis evaluation, and healthcare outcomes.

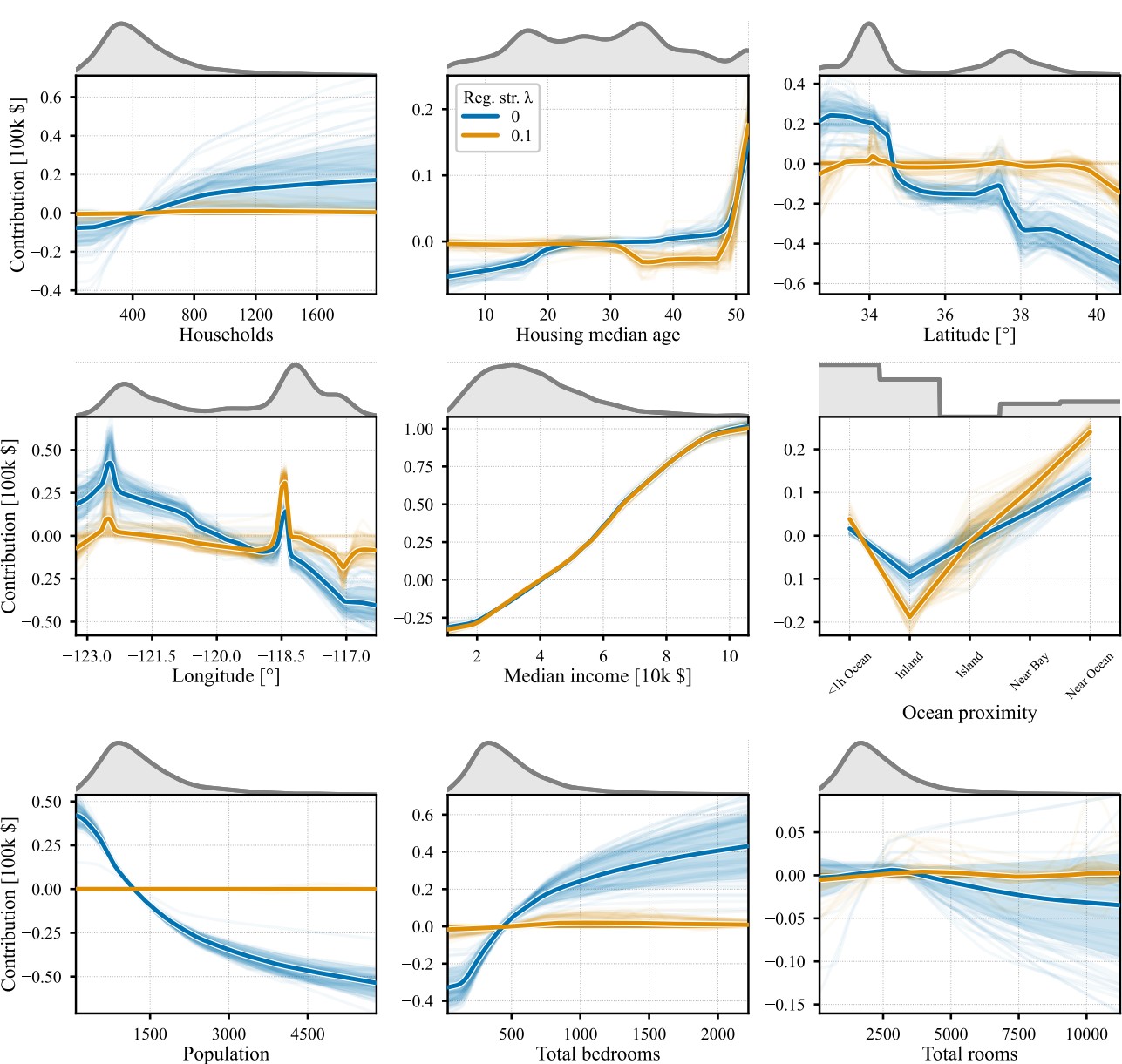

*Figure 14.* Shape functions of all input features in the case of the California Housing dataset. Depicted are the average as well as individual contributions with and without concurvity regularization using 60 initialization seeds each.

