# OpenReview forum: "Curve your Enthusiasm: Concurvity Regularization in Differentiable Generalized Additive Models"
_ICML.cc/2023/Workshop/IMLH — IMLH 2023 Poster_

### Official Review · Reviewer_AqB4 · 2023-06-12
**A regularizer for improving interpretability of GAMs is proposed**

**Rating:** 7
**Confidence:** 3

**Review:**

This work propose imposing a constraint on GAMs in order to remove concurvity and improve interpretability of these models. The proposed regularizer enforces "pairwise uncorrelatedness" which is sufficient for ruling out concurvity. However, this is not a necessary condition. Perhaps a less restrictive constraint results in a better models (in terms of performance)? It would be helpful if authors add a complexity discussion and discuss the computation cost of proposed method. Is there any other way of imposing another, perhaps computationally more expensive regularizer which enforce a milder constraint? In the experiment section, results are reported for both tabular and time series data (MIMIC and synthetic data), using Neural Additive Models and NeuralProphet models respectively. The experiments results are satisfactory.

---

### Official Review · Reviewer_rq2w · 2023-06-14
**great aspect to improve generalized additive models**

**Rating:** 7
**Confidence:** 3

**Review:**

This paper recognize and provided a great aspect to improve GAMs - their concurvity may hamper their interpretability. It proposes a simple yet effective regularizer to tackle this problem, which penalizes pairwise correlations of the non-linearly transformed feature variables. Experiments on both synthetic and real-world data shows that the proposed method can improve interpretability by reducing concurvity without much sacrifice on prediction performance.

The paper is well-written and the method is natural and straightforward. The experiments are well-organized, but it would be better if more numeric comparisons to the baseline are added, which can make the results clearer. Currently all of the results are visualized in figures.

---

### Meta-Review · Area_Chair_qmwz · 2023-06-18

**Recommendation:** Accept (Poster)
**Confidence:** 4

**Metareview:**

This work propose imposing a constraint on GAMs in order to remove concurvity and improve interpretability of these models.

All reviewers found the paper sound and interesting, presenting a timely contribution to interpretable ML methods for healthcare.

---

### Decision · Program_Chairs · 2023-06-20

Accept (Poster)